# PolypDB: A Curated Multi-Center Dataset for Development of AI Algorithms in Colonoscopy

## Abstract

Colonoscopy is the primary method for examination, detection, and removal of polyps. However, challenges such as variations among the endoscopists' skills, bowel quality preparation, and the complex nature of the large intestine contribute to high polyp miss-rate. These missed polyps can develop into cancer later, underscoring the importance of improving the detection methods. To address this gap of lack of publicly available, multi-center large and diverse datasets for developing automatic methods for polyp detection and segmentation, we introduce PolypDB, a large scale publicly available dataset that contains 3934 still polyp images and their corresponding ground truth from real colonoscopy videos. PolypDB comprises images from five modalities: Blue Light Imaging (BLI), Flexible Imaging Color Enhancement (FICE), Linked Color Imaging (LCI), Narrow Band Imaging (NBI), and White Light Imaging (WLI) from three medical centers in Norway, Sweden, and Vietnam. We provide a benchmark on each modality and center, including federated learning settings using popular segmentation and detection benchmarks. PolypDB is public and can be downloaded at `https://osf.io/xxxx/`. More information about the dataset, segmentation, detection, federated learning benchmark and train-test split can be found at `https://github.com/xxxxx/PolypDB`.

## 1 Introduction

Colorectal cancer (CRC) represents the third highest cancer incidence and is the second most common cause of cancer-related death worldwide. In 2020, approximately 1.9 million new cases of CRC were detected, causing approximately 935,000 deaths Sung et al. (2021). The relative five-year survival rate for persons younger than 64 years is 68.8% Yabroff et al. (2021). Colonoscopy is the gold standard for detecting CRC and removal of precancerous lesions such as polyps and very early CRCs. However, colonoscopy is an operator-dependent procedure causing a significant variation in polyp detection Hetzel et al. (2010). Smaller polyps, diminutive ($\leq 5$ mm) or (6 to 9 mm) sized colon polyps are often missed by the endoscopists. The adenoma miss-rate is reported to be 20%–24% Leufkens et al. (2012) and some missed polyps develop into CRC later on called postcolonoscopy CRC or interval cancer Rutter et al. (2018). For a couple of years, computer-aided detection (CADe) systems for polyp detection are commercially available and have shown to increase the adenoma detection rate but the polyps are just marked with a bounding box and do not help the endoscopists to delineate the polyp and confirm complete resection of the polyp, essential to avoid recurrence and potentially post colonoscopy CRC risks.

Precise delineation of polyps may be very helpful, especially for difficult ones, such as sessile serrated lesions (SSL). Accurate polyp segmentation is challenging because (i) polyp changes their characteristics over time during their development stage, (ii) their shape, size, colors, and appearance may be very similar to the surrounding mucosa, (iii) In some cases, there is a mucous covering the polyp acting as *camouflage* that might trick the endoscopists, even with state-of-the-art (SOTA) deep learning algorithms showing false positives, (iv) imaging device introducing artifacts such as blurriness, flares, and lighting conditions that also affect the colonoscopy procedure, for example, objects too close to the camera, under or over scene lighting, low resolution of capsular endoscopes, overexposure, reflection from the bright spot, low contrast areas and (v) the presence of surgical

Table 1: An overview of colon polyp datasets with a minimum of 1000 samples.

| Dataset | Findings | Size | Availability |
|---|---|---|---|
| Kvasir-SEG Jha et al. (2020) | Polyps | 1000 images[†] | open academic |
| HyperKvasir Borgli et al. (2020) | GI findings and polyps | 110,079 images and 374 videos | open academic |
| Kvasir-Capsule Smedsrud et al. (2021) | GI findings and polyps[◇] | 4,741,504 images | open academic |
| CVC-VideoClinicDB Bernal & Aymeric (2017) | Polyps | 11,954 images[†] | by request[•] |
| ASU-Mayo polyp database Tajbakhsh et al. (2015) | Polyps | 18,781 images[†] | by request[•] |
| BKAI-IGH Ngoc Lan et al. (2021) | Polyps | 1000 images[†] | open academic |
| PolypGen Ali et al. (2023) | Polyps | 1531 images[†] and 2000 video frames | open academic |
| **PolypDB (Ours)** | Polyps | 3934 polyp images from 3 centers | open academic |

[†]contains ground truth segmentation masks   [◇]Video capsule endoscopy   [•]Not available anymore

instruments and intestinal residue can also affect accurate polyp segmentation Jha et al. (2021). All these can affect colonoscopy procedures and limit accurate polyp segmentation and detection.

Fulfilling the gap between expert and non-expert endoscopists in detecting and diagnosing colon polyps is one of the most critical challenges in colonoscopy Ladabaum et al. (2013); Rees et al. (2017). Most of the DL methods perform reasonably well on the large adenomas ($\geq 10\,\mathrm{mm}$), which are easy to segment while overlooking small, diminutive, and even flat large SSLs, the main reason for right-sided post colonoscopy colorectal cancer van Toledo et al. (2022). However, SSLs are challenging to detect and delineate even for experienced endoscopists Van Rijn et al. (2006). Training DL algorithms on multi-center datasets can improve the generalizability and robustness of the network.

The main motivation of our work is to develop and publicly release a large-scale, multi-center polyp segmentation and detection dataset to be developed to support computer aided diagnosis (CAD) systems that are robust and generalizable for polyp segmentation and detection methods useful for integration into clinical settings. PolypDB consists of a diverse set of annotated images covering the global representativeness of the population and their annotations useful for performance evaluation and comparison of different Deep learning (DL) based algorithms. Our multi-center dataset consists of data from a variety of sources, imaging modalities (Blue laser imaging (BLI), Flexible spectral Imaging Color Enhancement (FICE), white light imaging (WLI), linked color imaging (LCI)), populations (Norway, Vietnam, Sweden), acquisition protocols (Fujinon system, Olympus) and imaging conditions captured by a multi-national expert that are better for early polyp diagnosis. Furthermore, we exploit this multi-center dataset and propose developing new benchmarks for polyp detection and segmentation both modality and center-wise. The main contributions of this work are as follows:

1. **PolypDB —** We present PolypDB, a multi-center, multi-modality polyp segmentation and detection dataset that consists of 3934 polyp images, pixel-precise ground truth and bounding box annotations collected from medical centers in Norway, Sweden and Vietnam. The diverse dataset helps the model enhance training and testing under real-world conditions.

2. **First-ever open access multi-modality dataset —** PolypDB consists of five distinct modalities such as BLI, FICE, LCI, NBI and WLI. This is the first-ever open-access dataset to feature five distinct modalities along with gastroenterologist-verified ground truth.

3. **Baseline benchmark —** We evaluated PolypDB on each modality using eight segmentation methods, five object detection methods, and six federated learning approaches, establishing a robust baseline benchmark.

## 2    POLYPDB DATASET DETAILS

### 2.1    STUDY DESIGN

PolypDB is a collection of colonoscopy examination images from three medical hospitals in Norway, Sweden, and Vietnam. Figure 1 presents the example images from different brands Fujifilm, Olympus, Pentax, primarily WLI, and with different types of digital staining such as BLI, FICE, LCI, and NBI, along with their corresponding bounding box ground truth and color coded segmentation masks. PolypDB is developed to address the urgent need for early detection and diagnosis of CRC precursors to reduce the incidence of CRC. Although some publicly available datasets exist (Table 1), there is no comprehensive modality-specific dataset to date. Also, multi-center open-access dataset is limited in the community. The multi-modality and multi-center dataset captures the

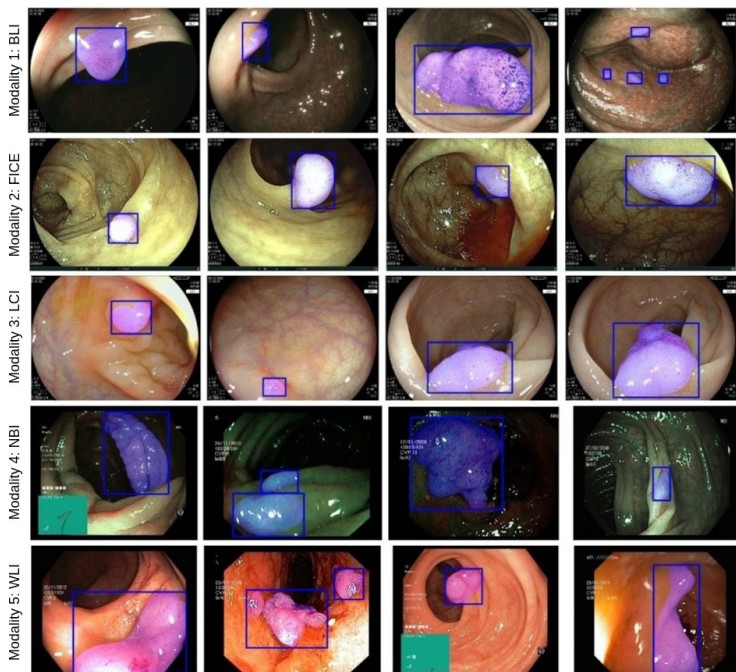

Figure 1: Examples of polyps in BLI, FICE, LCI, NBI, and WLI modalities from the PolypDB dataset, showcasing variations in shape, size, color, and appearance. Each image includes polyp bounding boxes and color-coded segmentation masks to show polyp ground truth.

regional and demographic disparities in CRC incidence rates, enhancing data diversity and broadening population representation. Additionally, having a multi-center dataset allows for the inclusion of different types of equipment and imaging protocols, which can also improve the robustness and generalizability of the CAD system, leading to better patient outcomes.

## 2.2 DATASET ACQUISITION: INCLUSION AND EXCLUSION CRITERIA OF THE COLONOSCOPY FRAMES

### 2.2.1 INCLUSION CRITERIA

The inclusion criteria for the polyp frames are as follows: Images with native colorectal polyp(s) in WLI mode and FICE mode, having a minimum resolution of $1280 \times 720$ pixels, and polyp's boundary must be clear and well-defined. Additionally, Boston Bowel Preparation Score (BBPS) $\geq 2$ and image should be captured in magnification mode.

### 2.2.2 EXCLUSION CRITERIA

The exclusion criteria ensure that we have high-quality and clinically relevant frames. We excluded frames where a polyp was resected (removed) or transported in a net, those with poor image quality, polyps injected with blue dye and snare around the polyp neck, and resection sites covered in blood, where residual polyps are unclear. Additionally, images were removed if it was unclear if a polyp or stool remnants, those showing normal anatomical structure or images in other image-enhanced modes (BLI, LCI), images in magnification mode. Images with poor quality, such as blurry, shaky, too dark, having a flare or having much liquid (feces, blood) and mucus, were removed. Images with already resected polyps or resection sites, images of polyps with submucosal injection, and images containing endoscopic tools such as caps, injection needles, snares, biopsy forceps, and clips were also excluded.

Table 2: **Data collection information for each center:** Data acquisition system and patient consenting information.

| Centers | System info. | Ethical approval | Patient consenting type |
| --- | --- | --- | --- |
| xxx, Norway | Olympus Evis Exera III, CF 190 | Exempted[†] | Not required |
| yyy, Sweden | Olympus Evis Exera III, CF 190 | Not required[‡] | Written informed consent |
| zzza, Vietnam | Fujinon system | Not required | Not required[‡] |
| zzzb, Vietnam | Fujinon system | Not required‡ | Not required[‡] |

[†] Approved by the data inspector. No further ethical approval was required as it did not interfere with patient treatment
[‡] Fully anonymized, no further ethical approval was required

## 2.3 DATASET COLLECTION AND CONSTRUCTION

### 2.3.1 XXX HOSPITAL, NORWAY (CENTER 1)

The polyp images were collected and verified by experienced gastroenterologists from *xxx* hospital trust in Norway. Some images have been collected from the unlabeled class of HyperKvasir dataset Borgli et al. (2020). There are 99,417 endoscopic frames in HyperKvasir dataset. We identified 3000 WLI polyps frames, labeled them and sent them to our gastroenterologists. Out of 3000 images, only 2588 were incorporated into our datasets. Others were excluded based on the exclusion criteria. Additionally, we selected 136 NBI images from the unlabeled HyperKvasir class. We curated the ground truth for both WLI and NBI, which was verified by a team of expert gastroenterologists. By labeling such datasets, we are making use of unlabeled frames, which were never explored for the development of new tools.

### 2.3.2 YYY UNIVERSITY HOSPITAL, SWEDEN (CENTER 2)

The images were collected and verified by an experienced gastroenterologist (10+ years of experience) from *yyy* Medical Hospital in Sweden. Although from their center, we received images from the entire GI tract, the number of polyp images was relatively limited. Based on exclusion criteria, we selected only 30 WLI polyp images and 10 NBI polyp images from *yyy* hospital. All these images were completely anonymized according to GDPR requirements for full anonymization.

### 2.3.3 ZZZA MEDICAL UNIVERSITY & INSTITUTE OF ZZZB, HANOI, VIETNAM (CENTER 3)

The dataset consisted of 1200 endoscopic images with polyps in 4 light modes: WLI, LCI, BLI and FICE. The data acquisition procedures for both centers are identical, and they examine similar populations. Therefore, we consider a single center in this study, given that both centers are located in the same city. Out of a total of 1200 images, 600 images were obtained from *zza*, while the other 600 images were sourced from *zzzb*. Specifically, *zzzz* consists of 1000 WLI polyp images, 60 LCI, 70 FICE and 70 BLI images. These images were labeled and annotated by three expert endoscopists with more than 10 years of experience. We provide both bounding box information and pixel-precise annotation for all images to make the dataset useful for both object detection and segmentation tasks. We also organized the dataset center-wise and modality-wise so that it could be useful to facilitate the research towards specific objectives in multiple directions.

## 2.4 ANNOTATION STRATEGIES AND QUALITY ASSURANCE

A team of 8 gastroenterologists (with most of them over 10 years of experience in colonoscopy) and one experienced senior researcher with a computer science background were involved in the data annotation, sorting, and the review process of the quality of annotations. The annotations were performed by a senior research associate who has extensive experience in data curation and development using online annotation tool called Labelbox https://labelbox.com/. All images were uploaded to Labelbox, and each frame was labeled considering the region of interest (area covered by polyp), and the ground truth for each sample was created. Each annotation was cross-verified by at least two senior gastroenterologists. Furthermore, we assign an independent reviewer (senior gastroenterologist) to review all 3934 images. All of the images were annotated by one researcher using the Wacom Cintiq tablet to minimize the heterogeneity in the manual delineation process.

During the review, the gastroenterologists marked if the frame represented colon polyps and should be included in the dataset. After that, they checked if the annotations for each polyp in a frame were "correct" and clinically acceptable. Finally, the non-polyp images were removed, and annotations were adjusted for incorrect annotations. For modality-wise organization, we provide "images", "corresponding ground truth masks" in the segmentation folder and "images" and "corresponding bounding box information" in the detection folder for each modality. The images and corresponding ground truth contain the same filename. For the center-wise data organization, we divide the dataset into three centers: *xxx*, *yyy*, and *zzz*. Each center has images, segmentation ground truth, and bounding box information useful for segmentation and polyp detection tasks. All images are encoded using JPEG compression.

## 2.5 ETHICAL AND PRIVACY ASPECTS OF THE DATA

The three medical hospitals involved in the PolypDB acquisition handled either all or at least two of the given steps, focusing on legal, ethical and privacy aspects of the dataset. Additionally, we believe releasing these datasets would help in the technological development, for example, the development of robust CAD system for polyps and there is a high potential benefit compared to the potential risk. Therefore, we make this dataset public after carefully considering ethical and privacy issues. Table 2 illustrates the ethical and legal processes fulfilled by each center, along with the endoscopy equipment and recorders used for the data collection. *1) Informed consent from the patient was obtained when required. Approval from the institution was always obtained. This also included the purpose of the study and how their datasets will be used. 2) Review and approval of the collected data from data inspectorate, institutional review board or local medical ethics committee depending on their country's regulations. 3) De-identification of the colonoscopy frame prior to the export from the hospitals' medical records release by following laws and regulations related to data privacy and protection in their nation.*

## 3 EXPERIMENTS AND RESULTS

### 3.1 DATASET AND IMPLEMENTATION DETAILS

**Dataset:** The experiments are conducted in two different settings: (i) modality-wise and (ii) center-wise. For modality-wise settings, we have 3558 WLI polyp images, 146 NBI images, 60 LCI images, 70 BLI, and 70 FICE images. We only experiment with WLI images for center-wise settings because it is common in all three centers. Although there are 136 NBI polyp images in center 1 and 10 polyp images in center 2, due to the minimal number of images present in both centers, we exclude them from the experiment.

**Implementation Details:** All experiments were conducted on a single NVIDIA RTX 3090. Datasets were split into $80\%$ training, $10\%$ validation, and $10\%$ testing. For polyp segmentation, images were resized to $512 \times 512$ and augmented with random rotations, flips, and coarse dropout. Models were trained for 200 epochs (batch size 12) using Adam ($1e^{-4}$) with binary cross-entropy + dice loss, early stopping, and *ReduceLROnPlateau*. For polyp detection, images were resized to $640 \times 480$ and augmented with flips, rotations, blur, mixup, mosaic, and cutmix. YOLO models were trained with AdamW ($1e^{-4}$), batch size 16, and uniform hyperparameters. For federated segmentation, models were trained under FedAvg across three centers using AdamW (weight decay 0.05, momentum 0.9), batch size 32, and 100 epochs. The learning rate started at 0.001 with cosine annealing (decay $\times 0.1$ every 30 epochs), and images were normalized per-center using local mean and standard deviation.

To evaluate the segmentation performance of the dataset, we employed several established segmentation methods.

### 3.1.1 SEGMENTATION RESULTS ON EACH MODALITY

Table 3 shows the results of different segmentation methods on each modality of the dataset.

**Results on BLI:** In the BLI dataset, DuAT emerged as the top-performing model, achieving the highest mIoU of 0.6979 and mDSC of 0.8048. DuAT also demonstrated high recall with a score of 0.9082 and maintained a high precision of 0.7647, resulting in the best F2 score of 0.8501.

Table 3: Comparison of quantitative results for segmentation on the PolypDB dataset. The highest and second highest scores are shown in **bold** and underline, respectively.

| Dataset | Method | mIoU | mDSC | Recall | Precision | F2 |
|---|---|---|---|---|---|---|
| PolypDB (BLI) | U-Net Ronneberger et al. (2015) | 0.1822 | 0.2855 | 0.6862 | 0.2180 | 0.3962 |
| | DeepLabV3+ Chen et al. (2018) | 0.6055 | 0.7293 | 0.8462 | 0.7146 | 0.7751 |
| | PraNet Fan et al. (2020) | 0.6581 | 0.7831 | **0.8876** | 0.7390 | 0.8348 |
| | CaraNet Lou et al. (2022) | 0.5853 | 0.7237 | 0.6895 | **0.8052** | 0.6978 |
| | TGANet Tomar et al. (2022) | 0.5217 | 0.6520 | 0.8108 | 0.6344 | 0.7076 |
| | PVT-CASCADE Rahman & Marculescu (2023) | 0.6737 | 0.7873 | 0.8750 | 0.7748 | 0.8205 |
| | DuAT Tang et al. (2023) | **0.6979** | **0.8048** | **0.9082** | 0.7647 | **0.8501** |
| | SSFormer-L Shi et al. (2022) | 0.6750 | 0.7848 | 0.8436 | 0.7708 | 0.8091 |
| PolypDB (FICE) | U-Net Ronneberger et al. (2015) | 0.1384 | 0.2021 | 0.5600 | 0.1425 | 0.2840 |
| | DeepLabV3+ Chen et al. (2018) | 0.6129 | 0.6759 | 0.6653 | **0.9441** | 0.6668 |
| | PraNet Fan et al. (2020) | 0.6013 | 0.6513 | 0.6559 | 0.7984 | 0.6530 |
| | CaraNet Lou et al. (2022) | 0.5694 | 0.6286 | 0.6082 | 0.8135 | 0.6146 |
| | TGANet Tomar et al. (2022) | 0.5922 | 0.6898 | 0.7086 | 0.7279 | 0.6960 |
| | PVT-CASCADE Rahman & Marculescu (2023) | 0.7209 | 0.7799 | 0.8110 | 0.7588 | 0.7971 |
| | DuAT Tang et al. (2023) | 0.5589 | 0.6746 | **0.9082** | 0.5867 | 0.7729 |
| | SSFormer-L Shi et al. (2022) | **0.7607** | **0.8300** | 0.8713 | 0.8013 | **0.8526** |
| PolypDB (LCI) | U-Net Ronneberger et al. (2015) | 0.3513 | 0.4712 | 0.5526 | 0.7644 | 0.4955 |
| | DeepLabV3+ Chen et al. (2018) | 0.8066 | 0.8898 | 0.8694 | 0.9294 | 0.8758 |
| | PraNet Fan et al. (2020) | 0.7936 | 0.8825 | 0.8890 | 0.8992 | 0.8834 |
| | CaraNet Lou et al. (2022) | 0.7600 | 0.8576 | 0.8335 | 0.9190 | 0.8398 |
| | TGANet Tomar et al. (2022) | 0.8358 | 0.9061 | 0.8816 | **0.9474** | 0.8899 |
| | PVT-CASCADE Rahman & Marculescu (2023) | 0.8344 | 0.9065 | 0.9074 | 0.9205 | 0.9056 |
| | DuAT Tang et al. (2023) | 0.8551 | 0.9194 | **0.9200** | 0.9247 | **0.9191** |
| | SSFormer-L Shi et al. (2022) | **0.8567** | **0.9207** | 0.9057 | 0.9466 | 0.9106 |
| PolypDB (NBI) | U-Net Ronneberger et al. (2015) | 0.2161 | 0.2986 | 0.6472 | 0.2622 | 0.3905 |
| | DeepLabV3+ Chen et al. (2018) | 0.6881 | 0.7733 | 0.8279 | 0.8511 | 0.7939 |
| | PraNet Fan et al. (2020) | 0.6749 | 0.7473 | 0.7816 | **0.8836** | 0.7618 |
| | CaraNet Lou et al. (2022) | 0.7249 | 0.8090 | 0.8312 | 0.8781 | 0.8194 |
| | TGANet Tomar et al. (2022) | 0.7317 | 0.8402 | 0.8368 | 0.8645 | 0.8354 |
| | PVT-CASCADE Rahman & Marculescu (2023) | **0.7769** | **0.8586** | **0.9385** | 0.8320 | **0.8941** |
| | DuAT Tang et al. (2023) | 0.7494 | 0.8260 | 0.8662 | 0.8741 | 0.8476 |
| | SSFormer-L Shi et al. (2022) | 0.7608 | 0.8432 | 0.9089 | 0.8462 | 0.8664 |
| PolypDB (WLI) | U-Net Ronneberger et al. (2015) | 0.7452 | 0.8250 | 0.8275 | 0.8936 | 0.8203 |
| | DeepLabV3+ Chen et al. (2018) | 0.8650 | 0.9168 | 0.9183 | 0.9380 | 0.9157 |
| | PraNet Fan et al. (2020) | 0.8570 | 0.9089 | 0.9046 | **0.9460** | 0.9042 |
| | CaraNet Lou et al. (2022) | 0.8582 | 0.9128 | 0.9149 | 0.9322 | 0.9114 |
| | TGANet Tomar et al. (2022) | 0.8536 | 0.9088 | 0.9165 | 0.9284 | 0.9104 |
| | PVT-CASCADE Rahman & Marculescu (2023) | 0.8731 | 0.9219 | 0.9268 | 0.9372 | 0.9227 |
| | DuAT Tang et al. (2023) | 0.8695 | 0.9197 | 0.9170 | 0.9437 | 0.9168 |
| | SSFormer-L Shi et al. (2022) | **0.8821** | **0.9294** | **0.9314** | 0.9438 | **0.9288** |

SSFormer-L followed closely with the second-highest mIoU of 0.6750, trailing by 2.29%. Both PVT-CASCADE and SSFormer-L provided close competition in mDSC, scoring 0.7873 and 0.7848, respectively. PraNet secured the second-best scores in recall (0.8876) and F2 (0.8348). Overall, DuAT demonstrated superior performance, showcasing its segmentation capabilities across multiple metrics.

**Results on FICE:** SSFormer-L demonstrated the best results in the FICE modality, achieving the highest mIoU of 0.7607 and mDSC of 0.8300, along with an impressive F2 score of 0.8526. Its recall score of 0.8713 was the second-best, while its precision score of 0.8013 remained competitive. PVT-CASCADE also performed well, with an mIoU of 0.7209 and mDSC of 0.7799. DuAT excelled in recall, achieving the highest score of 0.9082 for this modality, but its lower precision score of 0.5867 impacted its overall performance. Although DeepLabV3+ achieved the highest precision score of 0.9441, it did not lead in other metrics.

**Results on LCI:** For the LCI dataset, SSFormer-L once again led the performance metrics, achieving an mIoU of 0.8567 and an mDSC of 0.9207. It attained a high precision score of 0.9466 and an impressive F2 score of 0.9106, making it the top choice for LCI segmentation. DuAT also performed exceptionally well, with an mIoU of 0.8551 and mDSC of 0.9194, leading in recall with a score of 0.9200 and delivering a strong precision score of 0.9247. PVT-CASCADE closely followed, showing balanced results across all metrics, particularly in recall (0.9074) and precision (0.9205). While

TGANet exhibited a high precision of 0.9474, its slightly lower recall and mIoU scores prevented it from outperforming SSFormer-L and DuAT.

**Results on NBI:** In the NBI dataset, segmentation models exhibited varying performance levels. PVT-CASCADE, based on PVTv2-B2, demonstrated superior performance with an mIoU of 0.7769, mDSC of 0.8586, and recall of 0.9385, highlighting its efficacy in polyp identification. Additionally, it achieved an F2 score of 0.8941, underscoring its dominance in this domain. SSFormer-L followed with the second-best performance, achieving an mIoU of 0.7608 and mDSC of 0.8432, alongside a strong recall of 0.9089, which was 2.96% lower than that of PVT-CASCADE. PraNet secured the highest precision score at 0.8836. The DuAT model also delivered competitive results, particularly notable in recall (0.8662) and precision (0.8741), although it did not surpass the comprehensive performance of PVT-CASCADE.

**Results on WLI:** The WLI modality results were highly competitive, with SSFormer-L standing out as the top performer, achieving the best mIoU of 0.8821 and mDSC of 0.9294. SSFormer-L also led in recall with a score of 0.9314 and secured the second-best precision score of 0.9438, resulting in an impressive F2 score of 0.9288. PVT-CASCADE followed closely with an mIoU of 0.8731 and mDSC of 0.9219, demonstrating consistent performance with a recall of 0.9268 and precision of 0.9372. Although the performance gap between SSFormer-L and PVT-CASCADE was minimal, SSFormer-L's slight edge in multiple metrics made it the best choice for WLI segmentation. The DuAT model also delivered strong results, with a mIoU of 0.8695 and mDSC of 0.9197, showcasing competitive recall and precision scores.

Table 4: Quantitative detection results with previous methods. The highest and second highest scores are shown in **bold** and underline, respectively.

| Dataset | Method | mAP50 | mAP50–95 | mAP75 | Precision | Recall |
|---|---|---|---|---|---|---|
| PolypDB (BLI) | YOLOv8 Ultralytics (2023) | 0.659 | 0.502 | 0.559 | **1.000** | 0.318 |
| | YOLOv10 Wang et al. (2024a) | 0.534 | 0.416 | 0.485 | 0.840 | **0.500** |
| | YOLOv9 Wang et al. (2024b) | **0.688** | **0.558** | **0.638** | 0.846 | **0.500** |
| | YOLOv7 Wang et al. (2023) | 0.398 | 0.321 | 0.362 | 0.818 | 0.409 |
| | YOLOv5 Jocher (2020) | 0.618 | 0.499 | 0.534 | 0.899 | 0.404 |
| PolypDB (FICE) | YOLOv8 Ultralytics (2023) | 0.759 | 0.667 | 0.759 | 0.981 | 0.625 |
| | YOLOv10 Wang et al. (2024a) | **0.887** | **0.752** | **0.875** | **1.000** | **0.853** |
| | YOLOv9 Wang et al. (2024b) | 0.856 | 0.711 | 0.737 | 0.937 | 0.750 |
| | YOLOv7 Wang et al. (2023) | 0.734 | 0.642 | 0.734 | 0.856 | 0.750 |
| | YOLOv5 Jocher (2020) | 0.781 | 0.674 | 0.781 | 0.901 | 0.625 |
| PolypDB (LCI) | YOLOv8 Ultralytics (2023) | 0.833 | 0.771 | 0.833 | **1.000** | 0.667 |
| | YOLOv10 Wang et al. (2024a) | **0.995** | 0.831 | **0.995** | **1.000** | 0.854 |
| | YOLOv9 Wang et al. (2024b) | 0.972 | **0.878** | 0.972 | 0.857 | **1.000** |
| | YOLOv7 Wang et al. (2023) | 0.754 | 0.581 | 0.754 | 0.833 | 0.833 |
| | YOLOv5 Jocher (2020) | 0.833 | 0.687 | 0.833 | **1.000** | 0.667 |
| PolypDB (NBI) | YOLOv8 Ultralytics (2023) | 0.659 | 0.502 | 0.559 | **1.000** | 0.318 |
| | YOLOv10 Wang et al. (2024a) | 0.534 | 0.416 | 0.485 | 0.840 | **0.500** |
| | YOLOv9 Wang et al. (2024b) | **0.688** | **0.558** | **0.638** | 0.846 | **0.500** |
| | YOLOv7 Wang et al. (2023) | 0.398 | 0.321 | 0.362 | 0.818 | 0.409 |
| | YOLOv5 Jocher (2020) | 0.618 | 0.499 | 0.534 | 0.899 | 0.404 |
| PolypDB (WLI) | YOLOv8 Ultralytics (2023) | 0.913 | **0.766** | **0.868** | 0.883 | **0.880** |
| | YOLOv10 Wang et al. (2024a) | 0.555 | 0.391 | 0.434 | 0.603 | 0.525 |
| | YOLOv9 Wang et al. (2024b) | 0.912 | 0.757 | 0.836 | 0.899 | 0.856 |
| | YOLOv7 Wang et al. (2023) | 0.902 | 0.710 | 0.807 | **0.925** | 0.872 |
| | YOLOv5 Jocher (2020) | **0.916** | **0.766** | 0.852 | 0.918 | 0.872 |

## 3.2 DETECTION RESULTS ON EACH MODALITY OF THE DATASET

Table 4 illustrates the detection results on the PolypDB.

**Results on BLI:** In the BLI dataset, YOLOv9 achieves the highest performance with the best mAP50, mAP50-95, and mAP75 scores of 0.688, 0.558, and 0.638, respectively. In terms of precision, YOLOv8 surpasses other methods with a perfect score of 1.0000. However, for recall, both YOLOv9 and YOLOv10 achieve the same score of 0.5000. These results indicate strong performance in detecting positive cases, but the low recall scores highlight the challenge of improving the model's ability to predict positive cases and reduce missed detections in this dataset.

**Results on FICE:** The FICE dataset results showed YOLOv10 outperforming other methods with the best mAP50, mAP50-95, and mAP75 scores of 0.8870, 0.7520, and 0.8750, respectively. Additionally, YOLOv10 excelled in precision and recall, achieving scores of 1.000 and 0.8530, respectively, making it the most robust model for this modality.

**Results on LCI:** In the LCI dataset, YOLOv10 achieved the highest mAP50 and mAP75 scores of 0.9950, although YOLOv9 closely followed with the best mAP50-95 score of 0.8780. YOLOv10 also demonstrated superior performance in precision with a score of 1.000, while YOLOv9 achieved the best recall score of 1.0000, highlighting its effectiveness in identifying true positive cases.

**Results on NBI:** For the NBI dataset, YOLOv9 delivered the best results with a mAP50 score of 0.6880, a mAP50-95 score of 0.5580, and a mAP75 score of 0.638. YOLOv8 achieves the highest precision score with 1.0000, but the YOLOv10 and YOLOv9 have the highest recall scores, both achieving a score of 0.5000, showing their balanced performance in this modality. In addition, based on the dataset characteristics, and the benchmark results, we can assume that this dataset is more challenging for the detection model to focus on the important feature of the polyp, thus leading to the redundant features learning, and low recall results.

**Results on WLI:** For the WLI dataset, YOLOv5 achieved the highest mAP50 score of 0.9160, while YOLOv8 and YOLOv5 tied for the best mAP50-95 score of 0.7660. YOLOv8 also achieved the highest mAP75 score of 0.8680 and demonstrated strong precision with a score of 0.8830. Moreover, YOLOv8 excelled in the recall, achieving the top score of 0.8800, closely followed by YOLOv5 and YOLOv7. These results highlight the robustness of the dataset in guiding models to achieve high performance.

### 3.3 Federated Segmentation Results on WLI

Table 5 in Appendix A.1 presents the federated segmentation results. We apply the FedAvg algorithm McMahan et al. (2017) on three centers.

**Average Results:** On an average, SSFormer-L performed best with mIoU of 0.9214 and mDSC of 0.9550. It also achieved the highest precision and F2, with recall being the second best. SSFormer-L is followed closely by PVT-CASCADE and DuAT, which achieved very similar results. PVT-CASCADE excelled in the recall (0.9541), whereas DuAT reported the second-best mIoU and mDSC.

**Results on *zzz*:** Similar to the average results, SSFormer-L attained the best outcomes on center *zzz* with mIoU of 0.9426 and mDSC of 0.9696. It also performed superior in precision and secured the second-highest scores in terms of recall and F2. PVT-CASCADE and DuAT closely matched their performance, where the former reported the highest recall (0.9757) and F2 (0.9713), and the latter achieved the second-highest mIoU and mDSC.

**Results on *yyy*:** On the center *yyy* data, DuAT was ranked as the top-performing model with the best scores in three metrics, including mIoU (0.9396), mDSC (0.9683) and F2 (0.9578). SSFormer-L proved to be the second-best performing model. Although DeepLabV3+ and TGANet excelled in precision and recall, respectively, their comparatively lower performance in other metrics prevented them from ranking among the best-performing models.

**Results on *xxx*:** Showing consistently superior performance, SSFormer-L achieved the best outcomes in this case as well, with the highest mIoU, mDSC, precision and F2 of 0.9123, 0.9487, 0.9614, and 0.9439, respectively. The next best results are obtained using PVT-CASCADE, which are very similar to the DuAT outcomes.

## 4 Discussion

The quantitative results across the diverse datasets and modalities in PolypDB highlight the effectiveness of contemporary segmentation models, especially those utilizing advanced backbone architectures like PVTv2 and MiT-B4. The variation in performance observed across modalities—NBI, WLI, BLI, FICE, and LCI—emphasizes the complex challenges inherent in polyp segmentation, where selecting the appropriate model architecture is crucial for attaining superior performance. Regarding polyp detection, the results from the YOLO family of models are promising, highlighting the quality of our dataset in supporting these models. Additionally, benchmark results and visual-

izations from our test sets reveal the challenges associated with boundary information learning in polyp detection. Specifically, the model's performance on boundary detection and its ability to learn low-frequency features need to be improved to prevent it from missing small polyps or camouflaged objects where the patterns are similar to the background. We encourage further research to focus on solving this problem to improve polyp detection techniques.

### 4.1 IMPACT OF MULTI-MODALITY AND MULTI-CENTER DATA

One of the key strengths of PolypDB is its inclusion of data from five distinct imaging modalities—BLI, FICE, LCI, NBI, and WLI—collected from three different medical centers across Norway, Sweden, and Vietnam. This diversity is crucial in ensuring that models trained on PolypDB can generalize well across different clinical environments and patient populations. The inclusion of multi-center data helps mitigate the risk of overfitting to a specific type of imaging or patient demography, a common challenge in medical image analysis. As our results demonstrate, models trained and evaluated on this dataset show consistent performance across different modalities, suggesting that PolypDB can serve as a valuable resource for developing more universal and robust polyp detection and segmentation models.

### 4.2 SUPERIOR PERFORMANCE OF PVT-CASCADE AND SSFORMER-L

In this study, PVT-CASCADE and SSFormer-L consistently demonstrated top-tier performance across several metrics, including mIoU, mDSC, recall, and F2 scores. Particularly in the NBI and LCI modalities, PVT-CASCADE stood out with its highest mIoU (0.7769 and 0.8344, respectively) and mDSC (0.8586 and 0.9065, respectively). This can be attributed to the powerful feature extraction capabilities of the PVTv2-B2 backbone, which effectively captures both global and local contextual information necessary for accurate polyp segmentation.

### 4.3 IMPLICATIONS FOR CLINICAL APPLICATIONS

PolypDB enables researchers to develop more accurate and generalizable CAD systems that can assist gastroenterologists in detecting and segmenting polyps with higher precision. This can reduce polyp miss rates, which is critical in preventing CRC. Additionally, the modality-specific benchmarks provided in this study offer guidance on selecting the most appropriate models for different imaging modalities, potentially improving the overall quality of colonoscopy procedures. The success of FL algorithm experiments demonstrated the significance of FL in colonoscopy applications, where data privacy is critical. Robust federated CAD-based algorithms (for e.g, SSFormer-L) show strong potential to enhance clinical outcomes.

## 5 CONCLUSION

We introduced PolypDB, a multi-center and multi-modality polyp segmentation and detection dataset for advancing polyp detection and segmentation in colonoscopy. It comprises 3,934 polyp images from diverse imaging modalities and multiple medical centers and addresses the critical need for robust and generalizable data in developing CAD systems. The dataset's diversity, in terms of imaging modalities and geographical locations, ensures that models trained on PolypDB can perform effectively across a wide range of clinical settings, thereby enhancing their applicability in real-world scenarios. Our extensive benchmarking of SOTA segmentation and detection models, as well as the federated learning experiments in this domain, showing that privacy-preserving training across centers is feasible without substantial loss in accuracy. Furthermore, we evaluated the dataset under adversarial attack settings, demonstrating the importance of robustness analysis for safety-critical medical applications. Together, these experiments highlight PolypDB not only as a static dataset but as a comprehensive benchmark suite supporting multiple research directions, including segmentation, detection, domain generalization, and federated learning. In upcoming work, we aim to develop a comprehensive video dataset that captures the temporal dynamics of colonoscopy, as well as exploring robust domain adaptation techniques across devices and hospitals.

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

Table 5: Federated Segmentation results on WLI. Weights are aggregated equally. The best scores are shown in **bold**, whereas the second best score is underlined.

| Method | mIoU | mDSC | Recall | Precision | F2 |
|---|---|---|---|---|---|
| **Average** | | | | | |
| TransNetR Jha et al. (2024) | 0.8771 | 0.9218 | 0.9289 | 0.9400 | 0.9236 |
| U-Net Ronneberger et al. (2015) | 0.6362 | 0.7349 | 0.7929 | 0.7648 | 0.7562 |
| U-NeXt Valanarasu & Patel (2022) | 0.7049 | 0.7837 | 0.7977 | 0.8554 | 0.7853 |
| DeepLabV3+ Chen et al. (2018) | 0.9022 | 0.9423 | 0.9426 | 0.9539 | 0.9408 |
| PraNet Fan et al. (2020) | 0.9048 | 0.9438 | 0.9390 | 0.9593 | 0.9397 |
| CaraNet Lou et al. (2022) | 0.8941 | 0.9362 | 0.9324 | 0.9528 | 0.9325 |
| TGANet Tomar et al. (2022) | 0.8837 | 0.9290 | 0.9445 | 0.9299 | 0.9345 |
| PVT-CASCADE Rahman & Marculescu (2023) | 0.9105 | 0.9477 | **0.9541** | 0.9492 | 0.9504 |
| DuAT Tang et al. (2023) | 0.9109 | 0.9478 | 0.9485 | 0.9550 | 0.9465 |
| SSFormer-L Shi et al. (2022) | **0.9214** | **0.9550** | 0.9503 | **0.9642** | **0.9517** |
| **Center: zzz** | | | | | |
| TransNetR Jha et al. (2024) | 0.9199 | 0.9524 | 0.9598 | 0.9578 | 0.9561 |
| U-Net Ronneberger et al. (2015) | 0.8072 | 0.8714 | 0.8788 | 0.9193 | 0.8729 |
| U-NeXt Valanarasu & Patel (2022) | 0.8018 | 0.8661 | 0.8660 | 0.9087 | 0.8635 |
| DeepLabV3+ Chen et al. (2018) | 0.9358 | 0.9656 | 0.9663 | 0.9678 | 0.9656 |
| PraNet Fan et al. (2020) | 0.9331 | 0.9640 | 0.9669 | 0.9643 | 0.9654 |
| CaraNet Lou et al. (2022) | 0.9198 | 0.9537 | 0.9613 | 0.9563 | 0.9578 |
| TGANet Tomar et al. (2022) | 0.9311 | 0.9629 | 0.9608 | 0.9684 | 0.9612 |
| PVT-CASCADE Rahman & Marculescu (2023) | 0.9369 | 0.9659 | **0.9757** | 0.9592 | **0.9713** |
| DuAT Tang et al. (2023) | 0.9397 | 0.9679 | 0.9714 | 0.9661 | 0.9698 |
| SSFormer-L Shi et al. (2022) | **0.9426** | **0.9696** | 0.9726 | **0.9685** | 0.9711 |
| **Center: yyy** | | | | | |
| TransNetR Jha et al. (2024) | 0.8784 | 0.9308 | 0.8848 | 0.9899 | 0.9020 |
| U-Net Ronneberger et al. (2015) | 0.6511 | 0.7287 | 0.6607 | 0.9820 | 0.6830 |
| U-Next Valanarasu & Patel (2022) | 0.6711 | 0.7414 | 0.7173 | 0.9449 | 0.7255 |
| DeepLabV3+ Chen et al. (2018) | 0.9159 | 0.9547 | 0.9225 | **0.9916** | 0.9349 |
| PraNet Fan et al. (2020) | 0.8968 | 0.9428 | 0.9087 | 0.9847 | 0.9216 |
| CaraNet Lou et al. (2022) | 0.8921 | 0.9406 | 0.9100 | 0.9802 | 0.9213 |
| TGANet Tomar et al. (2022) | 0.8285 | 0.8959 | **0.9883** | 0.8340 | 0.9461 |
| PVT-CASCADE Rahman & Marculescu (2023) | 0.9006 | 0.9438 | 0.9148 | 0.9829 | 0.9255 |
| DuAT Tang et al. (2023) | **0.9396** | **0.9683** | 0.9510 | 0.9869 | **0.9578** |
| SSFormer-L Shi et al. (2022) | 0.9186 | 0.9555 | 0.9257 | 0.9915 | 0.9370 |
| **Center: xxx** | | | | | |
| TransNetR Jha et al. (2024) | 0.8593 | 0.9091 | 0.9173 | 0.9314 | 0.9108 |
| U-Net Ronneberger et al. (2015) | 0.5683 | 0.6798 | 0.7534 | 0.7074 | 0.7066 |
| U-NeXt Valanarasu & Patel (2022) | 0.6610 | 0.7465 | 0.7645 | 0.8342 | 0.7486 |
| DeepLabV3+ Chen et al. (2018) | 0.8876 | 0.9323 | 0.9326 | 0.9476 | 0.9302 |
| PraNet Fan et al. (2020) | 0.8934 | 0.9356 | 0.9282 | 0.9569 | 0.9296 |
| CaraNet Lou et al. (2022) | 0.8834 | 0.9289 | 0.9215 | 0.9499 | 0.9226 |
| TGANet Tomar et al. (2022) | 0.8662 | 0.9161 | 0.9385 | 0.9153 | 0.9244 |
| PVT-CASCADE Rahman & Marculescu (2023) | 0.8990 | 0.9399 | **0.9452** | 0.9442 | 0.9417 |
| DuAT Tang et al. (2023) | 0.8985 | 0.9391 | 0.9399 | 0.9489 | 0.9372 |
| SSFormer-L Shi et al. (2022) | **0.9123** | **0.9487** | 0.9416 | **0.9614** | **0.9439** |

Chien-Yao Wang, Alexey Bochkovskiy, and Hong-Yuan Mark Liao. Yolov7: Trainable bag-of-freebies sets new state-of-the-art for real-time object detectors. In *Proceedings of the IEEE/CVF conference on computer vision and pattern recognition*, pp. 7464–7475, 2023.

Chien-Yao Wang, I-Hau Yeh, and Hong-Yuan Mark Liao. Yolov9: Learning what you want to learn using programmable gradient information. *arXiv preprint arXiv:2402.13616*, 2024b.

K Robin Yabroff, Angela Mariotto, Florence Tangka, Jingxuan Zhao, Farhad Islami, Hyuna Sung, Recinda L Sherman, S Jane Henley, Ahmedin Jemal, and Elizabeth M Ward. Annual Report to the Nation on the Status of Cancer, Part 2: Patient Economic Burden Associated With Cancer Care. *JNCI: Journal of the National Cancer Institute*, 113(12):1670–1682, 2021.

# A APPENDIX

## A.1 IMPACT OF ADVERSARIAL ATTACK ON POLYPDB

We introduce results for the adversarial attack problem with the Fast Gradient Sign Method (FGSM) method to evaluate the robustness of deep learning models on the PolypDB dataset. By introducing small perturbations to the colonoscopy frames, we aim to expose vulnerabilities in the segmentation models. These attacks are significant in medical imaging as slight inaccuracies in the segmentation or detection results can lead to misdiagnosis. Thus, we evaluate the impact of FGSM on the baseline segmentation models on our PolypDB dataset.

Table 5 shows the comparison of the qualitative results for segmentation for FSGM attack on the PolypDB dataset. From the table, we can observe that models like UNet, DeepLabV3+, DuAT, and

TGANet, which performed well on the clean dataset, show substantial drops in evaluation metrics such as mIoU and DSC under adversarial conditions, demonstrating their susceptibility to perturbations. For example, UNet consistently achieved a low mIoU of 0.0485, 0.0557, 0.1057, 0.1487, and 0.0985, showing its resilience under challenging conditions.

However, models such as PVT-CASCADE and SSFormer-L achieve consistently high-performance metrics, for example, PVT-CASCADE obtained an mIoU of 0.6038 for WLI, 0.5153 for BLI, 0.4159 for FICE and 0.2560 for BLI. Similarly, SSFormer-L obtained a high mIoU of 0.5831 for WLI and 0.5476 for LCI. This underscores the resilience of these models even under challenging conditions.

Due to the high image diversity, multi-center nature and high image quality, these models still retained competitive performance. From here, we can conclude that PolypDB is not only a critical resource for benchmarking and advancing segmentation methodologies but also useful resource for developing segmentation and detection algorithms that can withstand real-world adversarial challenges such as adversarial vulnerabilities.

## A.2 QUALITATIVE RESULTS

**Polyp Segmentation:** Figure 2 visualizes our segmented results of different methods on our dataset.
**Polyp Detection:** Figure 3 presents the bounding boxes visualization across five different datasets

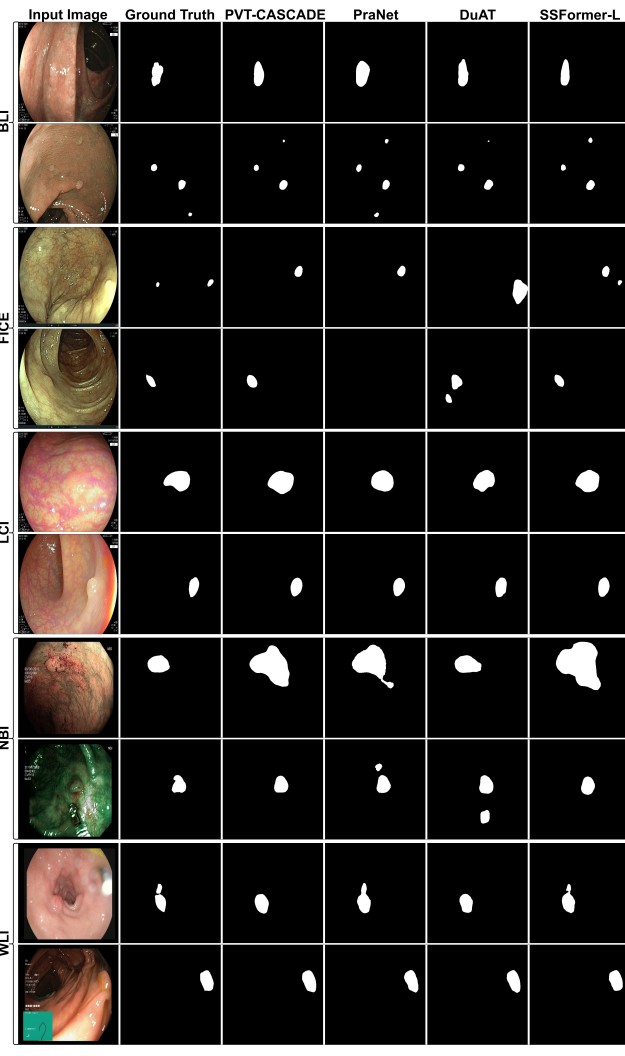

Figure 2: Qualitative results for the different methods across various modalities in the PolypDB dataset.

in all of the modalities of our dataset.

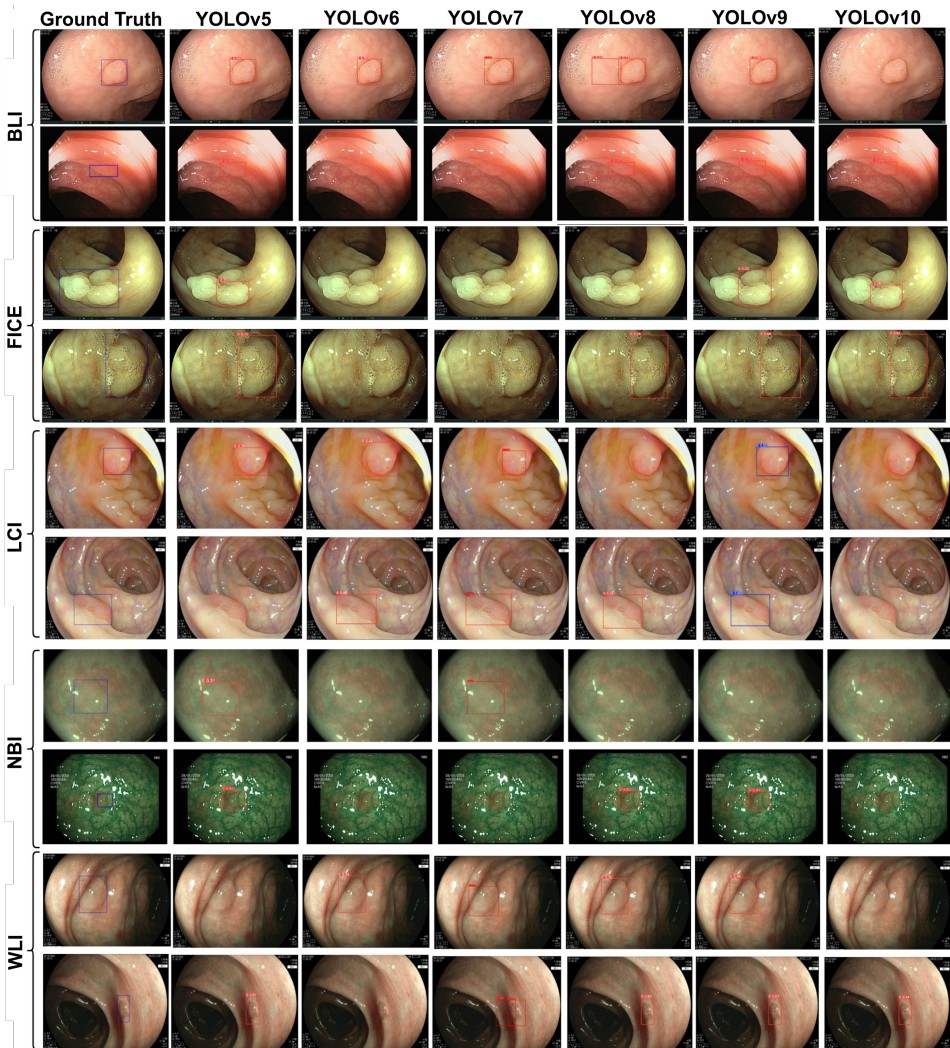

Figure 3: Qualitative results for the detection task on the different modalities in the PolypDB dataset.

