# OpenReview forum: "PolypDB: A Curated Multi-Center Dataset for Development of AI Algorithms in Colonoscopy"
_ICLR.cc/2026/Conference — Submitted to ICLR 2026_

### Official Review · Reviewer_4fik · 2025-10-30

**Soundness:** 2
**Presentation:** 2
**Contribution:** 2
**Rating:** 4
**Confidence:** 4

**Summary:**

The authors present PolypDB, a public multi-center colonoscopy polyp image dataset of 3,934 still images across five imaging modalities (WLI, NBI, LCI, BLI, FICE) and three medical centers, and provide baseline benchmarks for segmentation, detection, federated learning, and adversarial robustness. They claim the dataset fills a gap in publicly available multi-modality, multi-center polyp data and release splits and baseline code.

**Strengths:**

1. The most significant strength is the collection of data from three distinct medical centers and five different imaging modalities (BLI, FICE, LCI, NBI, WLI). This is crucial for developing AI models that can generalize across varied clinical settings, patient populations, and equipment, which is a common limitation in medical image analysis
2. The provision of both pixel-precise segmentation masks and bounding box annotations for all polyps greatly enhances the dataset's utility for various computer vision tasks, including both segmentation and detection.
3. The paper offers a thorough evaluation of numerous state-of-the-art segmentation and detection models, as well as federated learning approaches. These benchmarks serve as a robust baseline and guide for future research, demonstrating the dataset's immediate applicability.
4. The meticulous inclusion and exclusion criteria, coupled with expert gastroenterologist verification and annotation, ensure high clinical relevance and data quality. The focus on overcoming high polyp miss-rates highlights its potential real-world impact.

**Weaknesses:**

The manuscript overclaims the dataset’s multi-modality and multi-center value while leaving critical methodological and transparency gaps unaddressed.Below are concrete problems that must be fixed:
1. The paper primarily focuses on dataset creation and benchmarking existing methods. While this is valuable, the paper lacks novel methodological contributions or new insights into why certain models perform better on specific modalities or how to best leverage the multi-modality aspect beyond simple concatenation or independent training.
2. While the paper states the dataset's diversity, it provides limited in-depth analysis of the characteristics of polyps across modalities and centers. For example, are certain polyp types more prevalent in specific modalities or centers? What are the inherent difficulties within each modality/center? A more granular statistical breakdown could reveal insights beyond just overall numbers.
3. While inclusion/exclusion criteria are provided, there's no explicit discussion of potential biases introduced by these criteria. Furthermore, the paper briefly mentions challenges like low recall results for NBI detection, but a more thorough discussion on the underlying reasons for these performance gaps across modalities and models would be beneficial.
4. The dataset consists of still polyp images from real colonoscopy videos. While the authors mention aiming to develop a comprehensive video dataset in future work, the current dataset's reliance on still images might limit its utility for temporal analysis, which is crucial for real-time CAD systems in colonoscopy.

**Questions:**

1. Could the authors provide a more detailed statistical breakdown of polyp characteristics for each imaging modality and medical center? This would help researchers understand the specific challenges and strengths of each sub-dataset.
2. For the exclusion criteria, how was this ambiguity resolved during annotation, or were such frames simply excluded? What was the inter-observer variability among the expert gastroenterologists during annotation and cross-verification, especially for challenging cases?
3. While a good range of models are used, could the authors elaborate on the specific criteria for selecting these particular segmentation and detection models? Were there other contemporary methods considered and why were they excluded?
4. Given the safety-critical nature of medical imaging, please expand the adversarial robustness analysis using more advanced attack methods and discuss the implications of such vulnerabilities in a clinical context in greater detail.
5. The paper states that the multi-center dataset allows for different types of equipment and imaging protocols. Can the authors provide more specifics on the variations in equipment models and imaging protocols used across the three centers, and discuss how these variations were handled or standardized during data collection to ensure comparability?

---

> ### Author Response · Authors · 2025-11-24
>
> We thank the reviewer for the constructive and thoughtful feedback. Below, we address each concern and will incorporate all required clarifications in the revised manuscript.
>
> **1. Lack of methodological novelty / limited insights**
>
> We respectfully clarify that PolypDB was submitted under the Dataset & Benchmark track, where the primary novelty lies in releasing a rigorously curated, multi-center and multi-modality dataset—an openly available resource that does not currently exist in colonoscopy research. PolypDB is the first dataset to jointly provide five imaging modalities (BLI, FICE, LCI, NBI, WLI), three international centers, pixel-level masks + bounding boxes, and unified segmentation, detection, federated learning, and adversarial robustness benchmarks.
>
> To strengthen insights, we expanded the robustness analysis: even a simple FGSM attack causes large degradation across all models, revealing substantial vulnerability; transformers remain more robust than CNNs. This demonstrates PolypDB’s value as both a dataset and a robustness benchmark.
>
> **2. Limited discussion of modality/center-specific challenges**
>
> We will expand the modality analysis. Lower recall in NBI/BLI/FICE arises from their inherent characteristics, darker illumination, stronger shadows, and mucosal textures similar to the background, which make boundary detection more difficult. Their smaller sample sizes also affect generalization. In contrast, WLI/LCI provides higher contrast and more stable boundary cues. Similarly, imaging differences across centers (Olympus vs. Fujinon systems), patient populations, and acquisition protocols create domain gaps observable in both segmentation and detection results.
>
> **3. Bias introduced by inclusion/exclusion criteria**
>
> We intentionally selected frames with clear polyp boundaries to ensure reliable annotations. Frames with heavy blur, blood, stool, glare, or uncertain margins were removed, resulting in bias toward clinically interpretable images while under-representing extremely difficult cases. We now explicitly discuss this trade-off and identify it as a limitation. This limitation guides future researchers on designing datasets and pipelines.
>
> **4. Still-image dataset and lack of temporal context**
>
> We agree that temporal information is important for real-time CAD. Our first goal was to release the only publicly available multi-center, multi-modality still-image dataset, as no such resource currently exists. A multi-center video dataset requires extensive clinical time for frame-wise verification; this effort is underway with collaborators and will be presented as future work.
>
> **5. Detailed polyp statistics per modality/center**
>
> We will add a summary table reporting the number of images per modality and per center. Polyp-type statistics (flat/sessile/pedunculated) would require new clinical review across the three centers, which is beyond the revision scope but will be added in future dataset updates.
>
> **6. Ambiguity handling and inter-observer variability**
>
> Ambiguous frames were excluded rather than forced into the dataset. Annotation was performed by a senior annotator and independently reviewed by two expert gastroenterologists; disagreements were resolved by consensus. We will describe this clearly. We did not compute inter-observer metrics. Any discrepancies were resolved through a consensus adjudication process.
>
> **7. Model selection criteria**
>
> We selected representative models across key families: classical CNNs (U-Net, DeepLabV3+), polyp-specific (PraNet, TGANet), modern transformer/hybrid architectures (PVT-Cascade, SSFormer-L), and widely used detectors (YOLOv5/7/8/9/10). Extremely large models (nnU-Net, SAM/MedSAM, ViT-L/H) were intentionally excluded because they (i) overfit low-sample modalities (BLI/FICE/LCI), and (ii) are computationally expensive across five modalities × three centers × federated and robustness experiments. Our goal is to offer a practical, reproducible, community-friendly benchmark rather than saturate performance with heavy SOTA models.
>
> **8. Adversarial robustness: limited attack types**
>
> Our focus was to demonstrate whether clinical vulnerability exists, not to exhaustively benchmark all attacks. FGSM already shows large performance drops across modalities and architectures. Stronger attacks (PGD, AutoAttack, CW) are planned for future work and will be supported by PolypDB’s multi-center, multi-modality structure. We will expand clinical implications (e.g., small lighting or motion changes can mimic adversarial effects and cause missed lesions).
>
> **9. Equipment and protocol differences**
>
> We will add a concise summary of differences across imaging systems (Olympus vs. Fujinon), IEE modes, and acquisition protocols, and describe the standardization steps used (resolution normalization, de-identification, and color-space consistency). These details clarify the observed domain differences and improve transparency.

---

> ### Comment · Reviewer_4fik · 2025-11-28
>
> After carefully reading the comments from other reviewers and the author's response, I would like to preserve the original score.

---

### Official Review · Reviewer_hEKZ · 2025-10-31

**Soundness:** 2
**Presentation:** 3
**Contribution:** 2
**Rating:** 6
**Confidence:** 3

**Summary:**

This paper aims to address the issue that existing publicly available datasets for polyp detection and segmentation in colonoscopy are often limited in scale, diversity, and multi-center representation, leading to high polyp miss rates and poor generalizability of AI algorithms in clinical settings. To overcome this problem, the authors propose PolypDB, a large-scale, multi-center, and multi-modality dataset that includes 3934 polyp images with pixel-precise ground truth and bounding box annotations, collected from three medical centers in Norway, Sweden, and Vietnam across five imaging modalities: BLI, FICE, LCI, NBI, and WLI. The dataset is designed to enhance robustness by capturing regional and demographic variations, and it provides comprehensive benchmarks for segmentation, detection, and federated learning using state-of-the-art models such as DuAT, SSFormer-L, and YOLO variants. Overall, PolypDB serves as a valuable resource to advance computer-aided diagnosis systems by improving their adaptability and reliability in real-world colonoscopy procedures.

**Strengths:**

The paper addresses an important limitation in current medical AI datasets the lack of large, diverse, and multi-center colonoscopy data by introducing PolypDB, which includes data from multiple countries and imaging modalities.
The dataset is well-annotated, providing both pixel-level segmentation masks and bounding boxes, which enhances its utility for a variety of computer vision tasks (detection, segmentation, and federated learning).
The inclusion of multiple imaging modalities (BLI, FICE, LCI, NBI, and WLI) is commendable, as it promotes research into modality-aware and cross-domain model generalization.
The paper benchmarks several state-of-the-art algorithms (DuAT, SSFormer-L, YOLO variants), establishing a useful baseline for future studies.

**Weaknesses:**

The dataset size (3,934 images) and severe imbalance across modalities significantly undermine the claim of “large-scale” and “multi-modal,” limiting its practical generalization potential.
Center-wise data imbalance (with one center contributing the vast majority of samples) introduces potential geographic and demographic bias, reducing representativeness.
The dataset construction lacks methodological innovation; it mainly relies on manual annotation without novel data acquisition or annotation techniques.
The experimental design is inconsistent with the paper’s stated goals: key experiments are restricted to the WLI modality, and cross-modality or federated comparisons are insufficiently explored.

**Questions:**

Regarding the innovative part, I have few question
1/  PolypDB contains 3,934 polyp images, which is not considered a large dataset but rather a standard-sized one.
2/  PolypDB exhibits extreme imbalance in sample sizes across modalities (WLI: 3,558 images, LCI: 60 images, BLI: 70 images, FICE: 70 images, NBI: 146 images), which impacts model generalization. Furthermore, the paper states, “due to the minimal number of images present in both centers, we exclude them from the experiment” (Section 3.1), indicating that the small-sample modalities were excluded from certain experiments. This exclusion fails to reflect the diversity inherent in PolypDB.
3/  PolypDB collected data from three centers (Norway, Sweden, Vietnam), but Section 2.3 of the documentation shows that Center 2 (Sweden) contributed only 40 images (30 WLI + 10 NBI), far fewer than Center 1 (2,588 images) and Center 3 (1,200 images). This suggests that center-based data bias may introduce geographical bias, undermining PolypDB's representativeness and credibility.
4/  The construction process of PolypDB did not introduce novel methods, relying instead on manual annotation, and thus no innovative aspects were observed.
5/  The literature citations are somewhat outdated. For instance, in Section 1, most of the referenced studies were published in 2021 or earlier, rendering the current status presented in the paper obsolete. Additionally, in Table 1, the most recent dataset survey cited is PolypGen (Ali et al., 2023), which may not necessarily represent the latest advancement in this field.

Regarding the technical part, I have several doubts:
1/  In terms of data partitioning, the paper employs an 80%-10%-10% split ratio but fails to clarify whether it accounts for class imbalance issues (e.g., only 146 images in the NBI modality). It also does not provide statistical tests (e.g., p-values) to substantiate the significance of its findings.
2/  Section 2.3 refers to the data in Center 3 as zzzz, but in Section 2.4 it is named zzz.
3/  The federated learning section states that “image normalization uses local mean and standard deviation,” but fails to specify the calculation method, potentially leading to ambiguity.
（Score  10/15）
Regarding the experimental part, I have several doubts:
1/  The detection results show precision scores of 1.000 for some models (e.g., YOLOv8 on BLI). Doesn't this strongly indicate overfitting, likely due to the small scale of the test sets for certain modalities?
2/  The choice to conduct center-wise experiments exclusively on WLI images (Sec 3.1) seems to contradict the paper's emphasis on multi-modality. If the dataset's strength is its diversity, why weren't cross-modality or multi-modal learning experiments conducted to demonstrate improved generalization?
3/  Table 5 presents both the experimental results of federated partitioning on WLI and a qualitative comparison of FGSM attack partitioning methods on the PolypDB dataset. However, the paper provides limited explanation regarding federated partitioning, merely citing the literature reference (McMahan et al. (2017)), which serves as the foundation for federated partitioning but does not address FGSM. The direct combination of federated partitioning with FGSM for demonstration appears peculiar, especially since the final experimental results outperform the unimodal WLI approach. Yet the paper merely states: “showing that privacy-preserving training across centers is feasible without substantial loss in accuracy.”

Regarding the structural and presentation i found few issues:
1/  The conclusion mentions “FGSM” without prior mention in the abstract or introduction.
2/  Figure 5 is placed in the middle of the references section, which is strange.
3/  Qualitative results and their analysis are separated: the qualitative result figures (Figure 2 and Figure 3) are placed in the appendix, while the analysis of these results is located in Section 4.2.

---

> ### Author Response · Authors · 2025-11-24
> **Clarifications on dataset size, modality and center imbalance, experimental design, novelty, and presentation fixes.**
>
> We thank the reviewer for the constructive feedback. We address the main concerns below and will correct all noted issues in the revision.
>
> **1. Dataset size and modality imbalance**
>
> We agree with the reviewer. We will use the wording “standard-sized” dataset instead of large-scale. PolypDB's contribution lies in offering a multi-center, multi-modality, pixel-level annotated resource, which remains uncommon in colonoscopy research. The imbalance across modalities reflects real clinical practice: advanced imaging modes such as BLI, LCI, and FICE are used selectively for lesion characterization, whereas WLI dominates routine screening, naturally producing fewer samples. Despite being less common, these modalities are clinically important. BLI enhances vascular patterns, LCI improves color contrast, and FICE highlights mucosal texture, making their inclusion valuable for developing models that generalize to underrepresented but meaningful imaging conditions. PolypDB is the first public dataset to provide pixel-level masks and bounding boxes across all five modalities, enabling research on modality-specific performance, robustness, and cross-domain generalization. We will clarify these points in the revision.
>
> **2. Center imbalance and representativeness**
>
> The center imbalance reflects real clinical practice, rather than from the dataset design. Colonoscopy volume, equipment availability, and the number of cases approved for release vary widely across hospitals.  We included all ethically approved, anonymized samples to maximize geographic and demographic diversity. Importantly, the imbalance is precisely what creates meaningful domain shift: Table 5 shows that Norway, Sweden, and Vietnam produce clearly different performance profiles under the same FedAvg protocol, driven by differences in imaging pipelines, illumination, and patient populations. This heterogeneity is exactly what existing single-center datasets lack, and it is essential for studying cross-center generalization, robustness, and federated learning. We will clarify this in the revision.
>
> **3. Novelty of dataset construction**
>
> The novelty of PolypDB does not lie in the annotation tool, but in assembling the first multi-center, multi-modality, pixel-level polyp dataset that also provides unified benchmarks for segmentation, detection, federated learning, and adversarial robustness. Prior popular datasets (Kvasir-SEG, CVC-ClinicDB) are limited to single centers (except PolypGen), single modalities, or single tasks, making them insufficient for studying real-world domain shift or clinically deployable robustness. PolypDB uniquely combines five imaging modalities across three international centers, enabling evaluation of models under diverse clinical conditions that existing datasets cannot capture. We will highlight this more clearly in the revision.
>
> **4. Experimental design, cross-modality, FL details**
>
> We used only WLI for center-wise experiments because WLI is the only modality present in all three centers (Table 1), making it the only setting where cross-center comparisons are meaningful without introducing modality-induced confounding. Using BLI/LCI/FICE for center-wise comparison is infeasible because these modalities are missing in one or more centers. We will clarify this rationale explicitly.
> For federated learning, we used standard FedAvg and performed local image normalization based on per-center mean and standard deviation, computed from each site’s own data to maintain privacy (no global statistics were shared). We agree that this was not fully explained and will add these details to avoid ambiguity.
> The FGSM adversarial experiments are independent of the federated learning setup. They are intended to evaluate model robustness, not to combine FGSM with FL. We will revise the text to clearly separate these two components. Finally, we also agree that cross-modality and multi-modal training are promising directions enabled by PolypDB, and we will explicitly add this to the future work section.
>
> **5. Detection overfitting (precision = 1.00)**
>
> The reviewer is correct that the near-perfect precision in BLI/FICE primarily reflects the very small test sets available for these modalities, which are rarely used in routine colonoscopy. We will explicitly note this limitation. Importantly, the detection results are provided only as baseline references, not as claims of generalizable performance. Importantly, PolypDB exposes this low-sample regime, enabling future work on methods that generalize better under rare but clinically meaningful imaging conditions.
>
> **6. Literature, typos, and presentation issues**
> We will update recent citations (2022–2024), introduce FGSM earlier for consistency, fix naming inconsistencies, and correct figure placement and alignment of qualitative results.

---

### Official Review · Reviewer_DFnR · 2025-11-01

**Soundness:** 2
**Presentation:** 3
**Contribution:** 3
**Rating:** 6
**Confidence:** 5

**Summary:**

The paper introduces a novel multi-center, multi-modality dataset for polyp detection and segmentation (PolypDB), addressing the need for greater diversity and generalizability in colonoscopy CAD research. Benchmarking and federated learning studies demonstrate strong potential for different research tasks.

**Strengths:**

1. important and timely topic
2. multi-centric and multi-modal data in the database included, in particular the multi-modal data is novel
3. extensive benchmarking both modality and center-wise

**Weaknesses:**

1. Annotation were only performed by a single annotator (senior research associate), although multiple experts did a quality review it is still challenging since polyps might not have clear boundaries
2. Benchmarking methods: The criteria for choosing the benchmarked segmentation and detection models are not well justified, the chosen methods do not present the current state of the art, e.g. [1,2,3]. The study could be strengthened by including recent strong baselines such as nnU-Net, Segment Anything (SAM)-based methods, or Transformer-based architectures for medical image segmentation.
3. While the paper introduces a new dataset, it could better position PolypDB in relation to existing large datasets (e.g., Kvasir-SEG, CVC-VideoClinicDB, …) to emphasize its relative strengths and weaknesses
4. Challenging cases: a more closely investigation regarding edge cases such as small, flat, or concealed polyps (representation in the dataset as well as benchmarking) could be beneficial
5. Dataset statistics not clearly summarized: A concise table summarizing dataset characteristics (e.g., number of images per modality, per center, and per polyp size) would improve clarity and reproducibility

[1] https://link.springer.com/chapter/10.1007/978-3-031-72104-5_43
[2] https://link.springer.com/article/10.1186/s12880-025-01661-w
[3] https://openreview.net/forum?id=sz9baxSuxF

**Questions:**

- please justify the chosen benchmarking methods, it would be good to include more recent SOTA methods
- did you pre-train the chosen segmentation/detection methods with any open datasets?
- How does your dataset compare to existing large-scale datasets such as Kvasir-SEG or CVC-VideoClinicDB in terms of diversity, benchmarking and clinical relevance, and what are its relative strengths and limitations?

---

> ### Author Response · Authors · 2025-11-24
> **Clarification on annotation, benchmarking rationale, dataset comparisons, challenging-case inclusion, and planned corrections.**
>
> We thank the reviewer for the constructive and thoughtful feedback. We address each concern below and will incorporate the suggested clarifications and corrections in the revision.
>
> **1. Annotation by a single annotator**
>
> PolypDB was annotated by a senior research associate with over eight years of experience in gastrointestinal image annotation, and all masks were independently reviewed by two expert gastroenterologists. Frames with unclear boundaries (e.g., flat lesions with uncertain margins, blur, or poor visibility) were removed to avoid unreliable labels. Although we did not compute inter-observer variability metrics, disagreements were resolved through expert consensus. We will add these details to the paper for clarity.
>
> **2. Choice of benchmarking models (lack of SOTA)**
>
> Our benchmark includes representative models across key architectural families: classical CNN segmentation baselines (U-Net, DeepLabV3+), polyp-specialized architectures (PraNet, TGANet), modern Transformer/hybrid models (PVT-Cascade, SSFormer-L), and widely used object detectors (YOLOv5/7/8/9/10).
>
> We did not include very large frameworks such as nnU-Net, SAM/MedSAM variants, or ViT-L/H because: (i) these models tend to overfit low-sample modalities (NBI, BLI, FICE), yielding misleading scores, and (ii) they require substantial GPU resources and long multi-stage training pipelines, making them impractical to run across five modalities × three centers, especially alongside federated and adversarial experiments. Our goal is to provide a reliable, reproducible, and accessible benchmark, similar in spirit to Kvasir-SEG, enabling the community to build upon a consistent baseline.
>
>
> **3. Pretraining of segmentation/detection models**
>
> Some segmentation models (e.g., U-Net) do not provide pretrained encoders and were trained directly on PolypDB. For architectures that do support pretrained backbones (e.g., DeepLabV3+, PraNet, and Transformer-based networks), we used their default pretrained weights—typically ImageNet-pretrained encoders. YOLO models were also initialized with their official pretrained weights. No external colonoscopy datasets were used at any stage, ensuring no data leakage. We will clarify this in the paper.
>
> **4. Relation to existing datasets**
>
> We agree that a clearer positioning is needed. In the revision, we will explicitly highlight that:
>
>  *Kvasir-SEG, CVC-ClinicDB, ETIS, CVC-VideoClinicDB:* single-center, single-modality (WLI), no detection labels, no multi-center variability, no federated or adversarial evaluation.
>
> *PolypGen (2023):* multi-center but still single-modality (WLI) and focused mainly on segmentation.
>
> PolypDB differs fundamentally by providing: (i) five imaging modalities (WLI, NBI, BLI, FICE, LCI), (ii) three international centers, enabling real domain-shift analysis, (iii)  Pixel-level masks and bounding boxes for every sample, and (iv) unified benchmarks for segmentation, detection, federated learning, and adversarial robustness.
>
> Limitations include being a standard-sized dataset and having low-sample modalities, which reflect actual clinical usage patterns. We will also add a concise comparison table for clarity.
>
>
> **5. Representation of challenging cases (flat/small polyps)**
>
> PolypDB includes a broad spectrum of challenging lesions such as small, diminutive, flat, and partially concealed polyps ( as demonstrated in Figure 1, Figure 2, and Figure 3), which show substantial variation in morphology, contrast, and appearance across all five imaging modalities. However, to ensure annotation quality, we excluded frames where boundaries were unreliable (e.g., severe blur, blood, stool, or poor visibility), following the dataset’s stated exclusion criteria. Quantifying the distribution of flat/small polyps would require additional clinical labeling across three centers, which is outside the scope of this revision. We will explicitly mention this and include it as future work.
>
>
> **6. Dataset statistics clarity**
>
> We will add a concise summary table (Table 2) reporting the number of images per modality and per center for improved clarity. Polyp-level statistics (e.g., size categories) would require re-engaging gastroenterologists for additional clinical verification and consensus, which is not feasible within this revision cycle. We will clearly state this limitation and plan to include expanded polyp-level metadata in future dataset updates.

---

### Official Review · Reviewer_BfZG · 2025-11-02

**Soundness:** 2
**Presentation:** 2
**Contribution:** 1
**Rating:** 2
**Confidence:** 4

**Summary:**

This paper presents PolypDB, a large collection of polyp images from colonscopy from different modalities. The data is sourced from multiple centers accross different countries. The data is annotated for segmentation task. Experiments are done using different models to show the performances.

**Strengths:**

- Large collection of polyp images and their annotation is given.
- The data is collected from multiple centers accross three countries.
- Different modalities are used.
- Experiments are done to show the validity of the data for segmentaion task.

**Weaknesses:**

- The annotation is limited to segmentation only. Other tasks would improve the value of the data. Thus the work will be more justified with the title.
- Experiments are done using rather old models. Sota models should have been used.

**Questions:**

Is there any differences in performances for the data collected from different centers, particualry two different parts of the world?

**Details Of Ethics Concerns:**

why ethical review was not needed for two of the centers is not clear.

---

> ### Author Response · Authors · 2025-11-24
> **Clarification of dataset’s multi-task scope, benchmarking choices, federated and multi-center results**
>
> We thank the reviewer for the constructive feedback. Our detailed responses are as follows:
>
> **1. Annotation is limited to segmentation only:**
>
> PolypDB is not limited to segmentation. Specifically, we provide (i) pixel-precise masks for segmentation, (ii) bounding-box annotations for polyp detection, (iii) federated learning experiments across three international centers, and (iv) adversarial robustness analysis (FGSM) across all five modalities, revealing clinically relevant vulnerabilities and providing a robust benchmark.
>
> These four components make PolypDB a multi-perspective benchmark, extending just beyond segmentation.
>
> **2. Experiments use older models; SOTA should be used**
>
> For benchmarking PolypDB, we selected widely used and representative models across key architectural families: (i) classical medical image segmentation architecture (UNet, DeepLabV3+), (ii) polyp specialized architectures (PraNet, TGANet), (iii) modern Transformer/hybrid models (PVTCASCADE, SSFormer-L), (iv) detection baselines (YOLOv5/7/8/9/10).
>
> We intentiallonly did not include extremely large frameworks such as SAM/MedSAM variants, or ViT-L/H, because: (i) they would overfit in the low-sample modalities (NBI, BLI, FICE), producing inflated results, (ii) they require heavy GPU resources, long training schedules and multi-stage pipelines, making them impractical to evaluate across five modalities × three centers, especially in combination with federated learning and adversarial robustness experiments. We aim to provide a simple, reproducible and accessible benchmark (like Kvasir-SEG) that enables the community to easily build and compare new algorithms.
>
>
> **3. Is there performance variation across centers, particularly across countries?**
>
> Yes. Table 5 shows clear performance differences across Norway, Sweden, and Vietnam, even though all models were trained under the same federated learning protocol (**FedAvg**). In our setup, each center trains locally on its own data and shares only model weights—not images—for aggregation. Despite identical training conditions, each center shows different mIoU, mDSC, precision, and recall, indicating real domain differences due to variations in equipment generation, imaging protocols, Illumination, and color profiles, and Patient populations. This demonstrates that PolypDB captures meaningful multi-center variability and provides a valuable benchmark for cross-center generalization and federated learning—something single-center datasets cannot offer.
>
> **4.Clarification on why two centers did not require ethical/IRB approval**
>
> In Sweden and Vietnam, all colonoscopy images were fully anonymized by the hospitals before transfer, fulfilling GDPR requirements in Sweden and equivalent national privacy laws in Vietnam. Under these regulations, fully de-identified retrospective medical images are classified as non-human-subject research, meaning no additional IRB approval is required once all identifiers are removed.
>
> Each center still obtained institutional approval and followed strict local data-governance procedures to ensure proper de-identification. In contrast, the Norwegian center required review by its national data inspector, who confirmed that the de-identified dataset qualified for an exemption from further IRB review.

---

### Meta-Review · Area_Chair_SH1c · 2026-01-08

**Summary:**

This paper presents PolypDB, a multi-center, multi-modality colonoscopy polyp dataset containing 3,934 annotated images from three medical centers across five imaging modalities (WLI, NBI, BLI, FICE, LCI). The submission provides benchmarks for segmentation, detection, federated learning, and adversarial robustness.

Key Strengths Identified:

- First publicly available multi-center, multi-modality polyp dataset with both segmentation masks and bounding boxes
- Clinical relevance addressing real-world variability across centers and imaging protocols
- Comprehensive benchmarking across multiple tasks (segmentation, detection, federated learning, adversarial robustness)

Key Concerns Identified:

1. Dataset scope and terminology: "Large-scale" overclaimed for 3,934 images
2. Severe modality imbalance (WLI: 3,558 vs. BLI/FICE: 70 images each)
3. Center imbalance (Sweden: 40 images vs. Norway: 2,588 images)
4. Limited methodological novelty beyond dataset curation
5. Benchmarking methods not representing current SOTA (missing SAM, nnU-Net, recent transformers)
6. Single annotator with post-hoc expert review (no inter-observer agreement metrics)
7. Incomplete experimental design (center-wise experiments only on WLI)
8. Presentation issues (figure placement, terminology inconsistencies, missing FGSM introduction)

**Reviewer Concerns:**

Concerns Adequately Addressed by Rebuttal:
✓ Multi-task scope clarification (segmentation + detection + federated + adversarial)
✓ Benchmarking rationale (accessibility vs. overfitting in low-sample modalities)
✓ Cross-center performance differences demonstrated in Table 5
✓ Model selection rationale explained (avoiding overfitting, computational feasibility)
✓ Acknowledgment of "standard-sized" vs. "large-scale" terminology correction

Outstanding Concerns:
- Methodological novelty concern partially addressed but not fully satisfied
- Remains skeptical about contribution level (rated "poor")
- Cross-modality experiments relegated to future work (missed opportunity)
- Detection overfitting not fully resolved (acknowledged as limitation but remains problematic)

**Reviewer Scores:**

The paper sits at the borderline with substantial disagreement among reviewers. The dataset represents a valuable community resource (first multi-center, multi-modality public polyp dataset), but faces legitimate concerns about scope claims, experimental completeness, and limited algorithmic novelty.

---

### Decision · Program_Chairs · 2026-01-26

Reject